# Human newborns form musical predictions based on rhythmic but not melodic structure

Roberta Bianco[1,2*], Brigitta Tóth[3], Felix Bigand[1], Trinh Nguyen[1,4], István Sziller[5], Gábor P. Háden[3], István Winkler[3], Giacomo Novembre[1]

1 Neuroscience of Perception and Action Lab, Italian Institute of Technology, Rome, Italy, 2 Department of Translational Research and New Technologies in Medicine and Surgery, University of Pisa, Pisa, Italy, 3 Institute of Cognitive Neuroscience and Psychology, HUN-REN Research Centre for Natural Sciences, Budapest, Hungary, 4 Department of Developmental and Biological Psychology, University of Heidelberg, Heidelberg, Germany, 5 Division of Obstetrics and Gynaecology at DBC, Szent Imre University Teaching Hospital, Budapest, Hungary

* roberta.bianco@unipi.it

## Abstract

The ability to anticipate rhythmic and melodic structures in music is considered a fundamental human trait, present across all cultures and predating linguistic comprehension in human development. Yet, it remains unclear the extent to which this ability is already developed at birth. Here, we used temporal response functions to assess rhythmic and melodic neural encoding in newborns ($N=49$) exposed to classical monophonic musical pieces (real condition) and control stimuli with shuffled tones and inter-onset intervals (shuffled condition). We computationally quantified context-based rhythmic and melodic expectations and dissociated these high-level processes from low-level acoustic tracking, such as local changes in timing and pitch. We observed encoding of probabilistic *rhythmic* expectations only in response to real but not shuffled music. This proves newborns' ability to rely on rhythmic statistical regularities to generate musical expectations. We found no evidence for the tracking of *melodic* information, demonstrating a downweighting of this dimension compared to the rhythmic one. This study provides neurophysiological evidence that the capacity to track statistical regularities in music is present at birth and driven by rhythm. Melodic tracking, in contrast, may receive more weight through development with exposure to signals relevant to communication, such as speech and music.

## Introduction

Music is an increasingly compelling means for understanding the development of a wealth of neuro-cognitive processes, including those that support communication through sound patterns [1]. From the earliest stages of development, the human brain relies on multiple auditory cues to extract meaningful patterns—such as words or melodies—from the acoustic environment [2–4]. This process is facilitated by

**Data availability statement:** All data underlying the findings described in this manuscript are fully available without restriction. The EEG data and analysis code are publicly available from the Open Science Framework (OSF) at https://doi.org/10.17605/OSF.IO/K758D. The EEG data are shared in accordance with the Continuous-event Neural Data (CND) format standard. The corresponding musical stimuli are available in the same repository under the STIMULI folder. All data used to generate the figures are included as Supporting Information files (S1–S9 Data).

**Funding:** R.B. is funded by the European Union (MSCA, PHYLOMUSIC, 101064334, https://marie-sklodowska-curie-actions.ec.europa.eu/). G.N. and F.B. are funded by the European Research Council (ERC, MUSICOM, 948186, https://erc.europa.eu/homepage). T.N. is funded by the European Union (MSCA, SYNCON, 101105726, https://marie-sklodowska-curie-actions.ec.europa.eu/). B.T., G.P.H., and I.W. are funded by the Hungarian National Research Development and Innovation Office (ANN131305, FK139135, and K147135, respectively, https://nkfih.gov.hu/english-nkfih). The funders did not play any role in the study design, data collection and analysis, decision to publish, or preparation of the manuscript.

**Competing interests:** The authors have declared that no competing interests exist.

**Abbreviations:** ASR, Artefact Subspace Reconstruction; BERA, Brainstem Evoked Response Audiometry; CI, confidence intervals; EEG, electroencephalography; ERPs, event-related potentials; ICA, independent component analysis; IDyOM, information dynamic of music; IOI, inter-onset-interval; IPI, inter-pitch-interval; LMMs, linear mixed-effects models; mTRF, multivariate temporal response function; SE, standard errors;

the integration of sequential information and, thereby, by the extraction of statistical patterns along temporal and spectral dimensions, such as timing and pitch [5,6]. In music, tracking of statistical patterns is largely implicit [7], allowing the brain to anticipate events or patterns that occur more frequently than others based on both recent and past contexts. Expectations, therefore, build on statistical regularities acquired in real time as the current sequence unfolds, and/or retrieved from prior exposure. This process permits listeners to recognize rhythmic (temporal) and melodic (spectral) patterns [8], as well as to anticipate *when* an event will occur and *what* it will be [9–11]. Such rhythmic and melodic expectations are the backbone of music perception and appreciation [12] and are assumed to have contributed to the evolution and development of human musicality [13–18].

Based on cross-species studies, rhythmic and melodic expectations in primate species seem to have evolved along different phylogenetic pathways. Sensitivity to rhythmic patterns was observed in nonhuman primates, suggesting deep phylogenetic roots [19–24]. In contrast, the sensitivity to melodic patterns based on pitch relations appears more variable, if not absent, in nonhuman primates and may be unique to humans within the primate lineage [19,25–27]. This observation raises an important question: are humans naturally predisposed to melodic tracking? Answering this question is challenging yet important for understanding how biological predispositions, along with cultural traits, shape the complex spectrum of human musical abilities observed worldwide [28,29].

Here, we take human newborns as a testbed for studying the human brain's predisposition to process music, specifically its rhythmic and melodic aspects. Newborns' auditory responses can be reliably recorded using electroencephalography (EEG) [30,31], and these responses are marginally influenced by prior exposure compared with those measured at any later developmental stage (but see [32–36]). Compelling evidence suggests that the human brain engages with sounds already in utero, as fetuses discriminate, habituate to, and memorize sounds [37]. By approximately 35 weeks of gestation, fetuses begin to respond to music with changes in heart rates and body movements [38]. What remains unclear is which specific aspect of music—namely its rhythmic or melodic structure—drives these early predispositions.

In terms of rhythm perception, EEG studies demonstrated an early neural tuning to temporal structure in the human neonatal brain, such as specialization for temporal cues in both speech [39] and nonspeech signals [40], adaptation to the presentation rate of temporal patterns [41], tracking of meter-related frequencies [42], and perception of the beat [43]. Also, studies in newborns suggest that exposure to structured temporal input, such as music, can strengthen auditory networks and scaffold later language development [44,45]. Despite this evidence, it remains unclear whether newborns use rhythmic statistical regularities beyond sound periodicities, such as transition probabilities, to form temporal expectations [46]. In terms of melodic capacities, EEG studies showed that newborns exhibit discrimination of pitch independent of timbre [47] and detection of highly surprising events, such as deviants from deterministic patterns of tones [48] or regularities in sequences of tone intervals [49,50].

These studies provide preliminary evidence for expectations based on probabilistic distributions of melodic information. Yet they tested only the two tail-ends of such presumed probabilistic distribution: very frequent versus very infrequent events, ignoring the wide range of note-by-note surprises of real music. This leaves it unclear whether newborns can form melodic expectations whilst listening to continuous naturalistic music, as observed in adults [19,51,52]. Finally, because melodic and rhythmic abilities have often been studied separately, the weights of rhythmic and melodic expectations during music processing at birth are unknown.

Here, we investigate neural tracking of expectations based on both timing and pitch structures to understand how the newborn brain weights these musical features while listening to naturalistic musical stimuli (i.e., classical piano pieces). Therefore, unlike traditional paradigms, our design directly assesses rhythmic and melodic tracking within a full, ecologically valid stimulus, rather than inferring them from detection of salient irregular sounds. Rhythmic and melodic expectations can be generated through different anticipatory mechanisms sensitive to different features of the stimulus—from surface acoustical attributes to local and global event-based probabilities. Thus, using the multivariate Temporal Response Function analysis (mTRF) [53,54], we measured how multiple features of the continuous musical stimuli—namely 'low-level' acoustic features and 'high-level' probabilistic rhythmic/melodic information—predict human newborns' EEG responses to music. As in previous human and nonhuman primate work [19,51,52], we assessed neural encoding of J. S. Bach's piano monophonic pieces—rich musical stimuli combining both melodic and rhythmic probabilistic structures. Based on previous findings of rhythmic but not melodic tracking in nonhuman primates [19], we hypothesized that human newborns would show a similar pattern if these abilities were inherited phylogenetically. This would imply that whilst rhythm encoding is embedded in the human brain from the outset, melodic encoding might develop more slowly with experience and behavioral relevance. Conversely, if, unlike other nonhuman primates, rhythmic sensitivity and melodic sensitivity each emerge in parallel in humans, then human newborns might already exhibit some capacity for melodic encoding, potentially comparable to rhythmic encoding, as observed in adults [19,51].

## Results

An mTRF analysis was carried out to assess the neural encoding of musical expectations in humans at birth (Fig 1A). Newborns were exposed to musical melodies (real condition) and control stimuli (shuffled condition, where pitch and note timings were shuffled over time to create sequences with disrupted musical regularities). Musical melodies composed by Bach contain the regular melodic and rhythmic patterns typically found in tonal Western music. In contrast, the shuffled stimuli lack comparable predictability in pitch or timing (including a weak sense of musical beat), despite being acoustically similar (see S1 Fig; stimuli are available at https://doi.org/10.17605/OSF.IO/K758D).

To objectively assess the predictability of the experimental stimuli, probabilistic expectations were estimated based on the information-theoretic properties of the stimuli using a variable-order Markov model of statistical learning (i.e., information dynamic of music [IDyOM]; [55]). The model learns statistical patterns from sequences of discrete symbols representing different stimulus attributes, specifically concerning pitch and timing. It leverages observations from the past (long- and short-term) musical context, and it computes Shannon's surprise (S) and entropy (E) of each note in a melody associated with pitch (Sp and Ep, respectively) and onset timing (St and Et). Surprise and entropy provide complementary characterizations of predictive processing: entropy captures the inherent uncertainty of an event, whereas surprise reflects the unexpectedness of that event given prior context. Including both predictors allows us to fully represent pitch and timing tracking, ensuring that we capture neural activity related to both aspects of musical anticipation. The estimates provided by the model confirmed that shuffled melodies were overall more unexpected than real melodies, both with respect to pitch (Sp: $W = 40$, $p = .002$; Ep: $W = 40$, $p = .002$; see Methods 'Statistical analysis' for details) and to timing (St: $W = 35$, $p = .036$; Et: $W = 33$, $p = .075$) (Fig 1B).

We further investigated the relationship between stimulus predictability and low-level acoustic features. Note-specific surprise and entropy estimates positively correlated with low-level acoustic features such as the magnitude of the latest

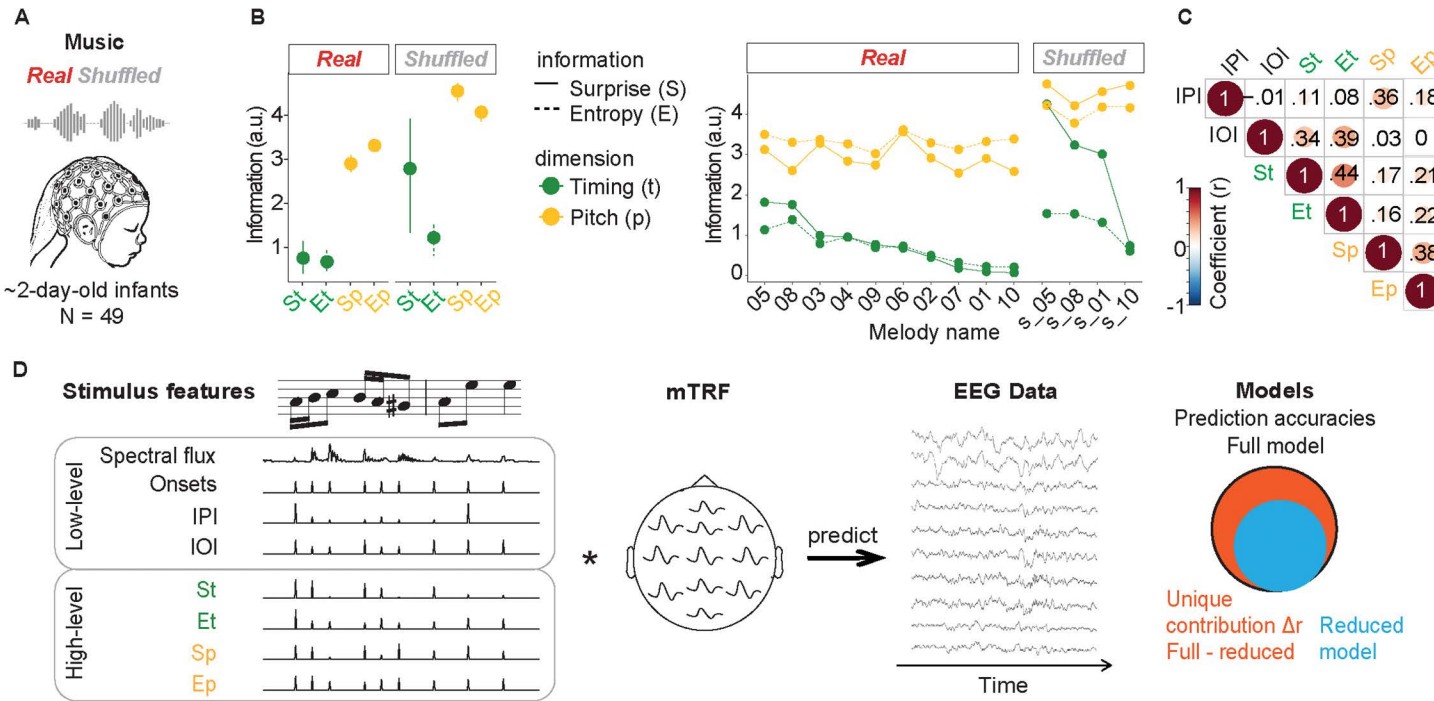

**Fig 1. Materials and methods. (A)** Experimental paradigm. We analyzed EEG data recorded from 49 sleeping human newborns while being exposed to monophonic piano melodies composed by J. S. Bach (real condition) and control stimuli (shuffled condition). **(B)** Surprise and entropy. Surprise and entropy associated with each note's timing (green, St and Et, respectively) and pitch (yellow, Sp and Ep, respectively) were estimated using an unsupervised statistical learning model trained on all stimuli. Dot plots display mean surprise and entropy associated with real and shuffled music, averaged across melodies (left panel), and separately for each melody (right panel). Error bars represent bootstrapped 95% confidence intervals (CI). See S1 Data. **(C)** Correlations between stimulus features. Pearson's correlations (*r* values) between the stimulus features: inter-pitch-interval (IPI), inter-onset-interval (IOI), and surprise and entropy associated with timing (St and Et) and pitch (Sp and Ep). See S1 Data. **(D)** Analytical approach. Multivariate Temporal Response Function (mTRF) models were fit to describe the forward relationship between multiple stimulus features and the EEG signal. The full TRF model (leftmost panel) included acoustic low-level features (spectral flux, acoustic onset, IOI, and IPI) and high-level features (surprise and entropy of pitch and timing). To assess the unique contribution of each feature (or set of features) to the EEG data, we run reduced models encompassing all variables but with the specified one being randomized in time (yet preserving the note onset times). We then calculated the difference in EEG prediction accuracy (Pearson's correlations, *r*) between the reduced models and the full model (Δr). On the rightmost panel, the light blue circle denotes information of a reduced model, with the variable(s) of interest being randomized. The orange area indicates the unique contribution of the variable of interest that leads to an increase in the explanatory power of the full model (black circle).

pitch or timing interval (i.e., inter-pitch-interval, IPI, or inter-onset-interval, IOI) (Fig 1C). Hence, in the following analyses, we assessed the unique contribution of probabilistic pitch and timing expectations, above and beyond the contribution of low-level acoustic processing (including IPI, IOI, acoustic onsets, and spectral flux) (Fig 1D). To do so, we derived single-participant TRFs by fitting multivariate lagged regression models (Fig 2A). We then estimated prediction accuracy (Pearson's correlations, *r*) between the EEG signals predicted by the TRF models and the actual EEG data (averaged across all participants, 'ground-truth EEG data' see Methods 'TRF analysis'), separately for each melody and EEG channel, using leave-one-melody-out cross-validation over a lag window ranging from −50 to 400 ms. To assess the unique contributions of the variables of interest to the EEG data, we trained reduced models that included a veridical representation of all variables except for the variable of interest, which was randomized (see Methods 'TRF Analysis'). Finally, we calculated the difference in prediction accuracy (Δr) between the reduced and full models, selecting the top 25% of channels with the highest prediction accuracy in the full model across conditions (see Methods for ROI definition).

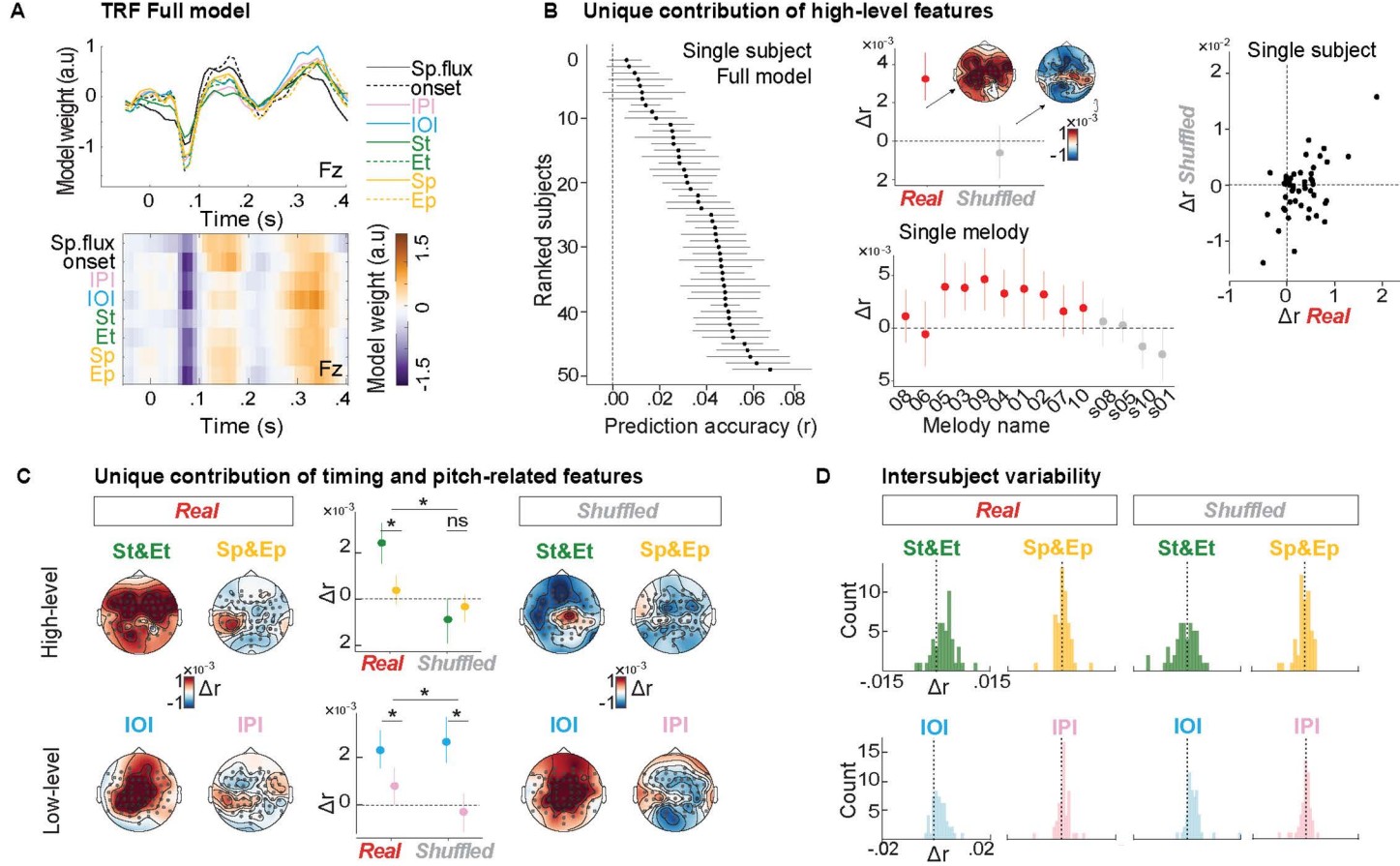

**Fig 2. Neural encoding of timing but not pitch expectations at birth. (A)** TRF full model. Ridge regression weights in time yielded by TRF for all predictors of the full model at the electrode Fz. Orange and purple colors indicate positive and negative weights, respectively. Zero on the time axis represents the note onset. **(B)** Unique contribution of high-level musical features. Left panel: black dots indicate EEG prediction accuracy of the full model of each infant, computed across 25% of channels with the highest prediction accuracy across conditions (best channels per infant, used in all following plots). Infants are ranked according to the prediction accuracy (r) of the full model. Central upper panel: group-average ($n = 49$) Δr resulting from the difference between full and reduced models assessing the unique contribution of high-level musical features (St, Et, Sp, and Ep) computed across each infant's best channels and plotted separately for real (red) and shuffled (gray) melodies with associated topographical maps. Central lower panel: grand-average Δrs are displayed for each melody, ranked from slower to faster tempo (see S1 Table). Error bars represent bootstrapped 95% CI. Right panel: scatter plot representing the relationship between Δr yielded by high-level musical features associated with real (x-axis) and shuffled (y-axis) melodies (each dot represents one participant). See S2 Data **(C)** Unique contribution of timing and pitch-related features. Topographical maps representing group-average Δr resulting from the difference between full and reduced models separately assessing the unique contribution of high-level timing features (St and Et), high-level pitch-related features (Sp and Ep), as well as low-level timing (IOI) and pitch-related (IPI) features across conditions (left: real; right: shuffled). Dot plots represent the group-average mean Δrs for the four reduced models (dots are color coded consistently with the colors used to label the high- and low-level musical features), computed across each infant's best channels. Error bars represent bootstrapped 95% CI; asterisks indicate the presence of significant main effects and interactions; 'ns' indicates nonsignificant effects. See S2 Data. **(D)** Intersubject variability. Histograms of individual Δrs for models reduced by St and Et, Sp and Ep, IOI, as well as IPI, separately for the two conditions.

## Encoding of probabilistic expectations in real but not shuffled music

Fig 2A shows the weights (see Methods 'TRF analysis') in time yielded by TRF for all stimulus features of the full model. Fig 2B (left panel) shows that the full model—including all features—predicted the EEG data with reasonable accuracy across virtually all participants, yielding correlation values comparable to those reported in previous TRF studies [51,56]. However, the intersubject variability was substantial, but not explained by the gestational age [57] (Spearman $\rho = 0.183$,

PLOS Biology

$p = 0.207$), perhaps reflecting the limited variability of this measure in our sample (gestational age mean $279.8 \pm 6.8$ GG, range 257–290 GG).

To what extent do probabilistic (high-level) expectations contribute to the neural signal? We tested the unique contribution of probabilistic (high-level) features derived from the IDyOM model (St, Et, Sp, and Ep) beyond low-level stimulus features (onset, spectral flux, IOI, and IPI). We thus compared the change in prediction accuracy ($\Delta r$) between the full model and the reduced model—where event-related predictors (St, Et, Sp, and Ep) were randomized both in real and shuffled music (Fig 2B, top central panel). A linear mixed effect model (see Methods 'Statistical analysis' for details) with the fixed factor Condition (real versus shuffled) yielded a main effect of Condition ($\chi^2(1) = 12.065$, $p < .001$) indicating encoding of probabilistic expectations in real but not shuffled music (real > shuffled: $b = .004$, SE $= .001$, $p = .005$; real > 0: $b = 0.003$, SE $= .0007$, $p < .001$; shuffled > 0: $p = .594$). These effects were not driven by any specific melody (Fig 2B, bottom central panel) and exhibited high variability across subjects (Fig 2B, right panel, see also S2A Fig for a visualization of the condition effect on $\Delta r$ values across individual participants and electrodes). This analysis demonstrates that the predictable structure of real (but not shuffled) melodies allows newborns to generate musical expectations over and above mere acoustic tracking.

**Timing- but not pitch-related expectations**

We tested whether the encoding of probabilistic expectations was specifically driven by pitch or timing structures (Figs 2C, 2D and S2B upper panels). We thus examined the difference between the full model and a reduced model, in which either St and Et (timing probabilistic TRF model) or Sp and Ep (pitch probabilistic TRF model) were randomized. A linear mixed effect model with fixed factor Condition (real versus shuffled) and TRF model (St and Et versus Sp and Ep) yielded a main effect of condition ($\chi^2(1) = 12.353$, $p < .001$) and an interaction between Condition and TRF model ($\chi^2(1) = 9.897$, $p = .002$). For real music, paired contrasts indicated encoding of probabilistic expectations based on timing but not pitch structure (St and Et > Sp and Ep: $b = .002$, SE $= 0.0004$, $p < .001$; St and Et > 0: $b = 0.0024$, SE $= .0005$, $p < .001$; Sp and Ep > 0: $p = .384$), whereas for shuffled music, neither of the two dimensions yielded significant effects (St and Et > Sp and Ep: $p = .856$; St and Et > 0: $p = .166$; Sp and Ep > 0: $p = .601$). This analysis demonstrates that newborns track the predictable rhythmic structure of the real melodies to generate expectations. In contrast, pitch-based probabilistic expectations do not appear to emerge with statistical significance.

As a control, we ran similar analyses to test the unique contribution of expectations driven by just immediate local changes in timing and pitch, as estimated by IOI and IPI (Figs 2C, 2D and S2B lower panels). We thus examined the difference between the full model and a reduced model, in which either IOI or IPI was randomized. A linear mixed effect model with fixed factor Condition (real versus shuffled) and TRF model (IOI versus IPI) yielded a main effect of TRF model ($\chi^2(1) = 48.225$, $p < .001$) and an interaction between Condition and TRF model ($\chi^2(1) = 5.032$, $p = .025$). Paired contrasts indicated encoding of IOI but not IPI for both real (IOI > IPI: $b = .001$, SE $= 0.0003$, $p = .001$; IOI > 0: $b = 0.0022$, SE $= .0005$, $p < .001$; IPI > 0: $p = .12$) and shuffled music (IOI > IPI: $b = .003$, SE $= 0.0005$, $p < .001$; IOI > 0: $b = .00026$, SE $= .0007$, $p = .002$; IPI > 0: $p = .705$). This analysis demonstrates that (low-level) expectations based on local temporal intervals are not altered by the rhythmic structure of the music, as IOIs were similarly tracked in real and shuffled melodies. It also shows that encoding of the pitch information did not reach significance in either condition (although IPI tracking approached significance when compared to zero in the real condition). Hence, the current results do not support the tracking of either pitch probabilistic expectations or local pitch change.

Note that notes carrying high surprise are often preceded by relatively larger IOIs, and this bias was stronger in real than in shuffled music (S3A Fig). However, the stronger tracking of probabilistic rhythmic expectations (St and Et model) in real than shuffled music cannot be explained by low-level timing alone, as St and Et regressors captured additional EEG variance beyond that explained by the preceding IOI (Fig 2C). We also conducted further control analyses. First, because IOIs were occasionally short, one could argue that consecutive ERPs may have overlapped, potentially confounding the

TRF results across conditions. This concern had already been addressed in our main analysis, where the IOI preceding each event was included as a regressor in the TRF model. To further rule out this possibility, we repeated the analysis, also adding the subsequent IOI as a regressor. The results of this analysis confirmed the findings reported above (S3B Fig). Second, to rule out the possibility that some, even if not all, infants were able to generate pitch-based probabilistic expectations, we explored whether those infants generating relatively stronger predictions for timing were also generating stronger predictions for pitch. To do so, we correlated $\Delta r$ across Sp and St but found no significant correlation for either real (Spearman $\rho = 0.234$, $p = 0.105$) or shuffled music (Spearman $\rho = 0.209$, $p = 0.15$).

### Converging evidence from Event-Related Potentials (ERPs)

To ground the TRF results in more widely used neurophysiological responses, we examined ERP responses to a subset of musical notes, specifically those carrying the highest and lowest 20% quantiles of surprise values (High S and Low S, respectively), separately for pitch and timing (Fig 3A). The ERPs consisted of a first negative peak (termed N1) followed by two broad positive-going deflections (P1 and P2) separated by a small (second) negative-going deflection (N2). The ERP waveforms resemble those previously observed in newborns evoked by auditory stimuli [58]. Furthermore, the waveform is reminiscent of TRF's regression weights (Fig 2A), suggesting that the TRF analysis primarily captured phase-locked auditory responses, as observed in previous studies [19,51,59].

Notably, the amplitude of the two positive-going deflections was enhanced in response to temporally unexpected (High S) compared to expected (Low S) notes, reaching significance in the second peak (from +.24 to +.37 s). This was observed for real but not for shuffled music. Conversely, no significant amplitude modulation was evoked by notes with unexpected pitch. This dissociation, together with the observation that pitch- and time-related surprise values (Sp and St) are weakly correlated (rho = .17), suggests that unexpected pitch and timing events are processed independently. These results fully align with the TRF results, confirming that newborns generate expectations based on the rhythmic rather than melodic structure of the musical stimuli. They further provide insights into a neurophysiological response, specifically a late EEG positivity, whose amplitude varies as a function of timing- but not pitch-related surprise. Together, these results warrant comparison with findings from previous studies that exposed human adults (Fig 3B) and Rhesus monkeys (*Macaca mulatta*, Fig 3C) to the same stimuli [19,51] to be further elaborated upon in the Discussion.

## Discussion

We employed continuous music stimuli with a rich melodic and rhythmic "alphabet" as a testbed for investigating the neurophysiology of music encoding in newborns. We demonstrated the feasibility of using naturalistic complex stimuli, such as Western tonal music, to examine different levels of auditory processing at birth. By determining how newborns use statistical regularities in melodic and rhythmic information to process music, our findings provide key contributions to understanding auditory development and its built-in biological constraints. Specifically, while rhythmic statistical regularities embedded in musical stimuli are neurally encoded already at birth, pitch-based information does not receive the same depth of processing, whether at low or high levels of encoding. This suggests that rhythmic and melodic sensitivities do not emerge in parallel in humans, with rhythm developing earlier than melody.

TRF analyses revealed high inter-individual variability in overall neural tracking of musical stimuli (Fig 2B left panel, see also [60]), likely stemming from the high variability in morphology and latency of newborns' auditory ERPs [61], and compatible with the notion that TRFs capture ERP-like responses [54]. Crucially, we showed that newborns track note-by-note predictability in real but not shuffled music, and that the rhythmic, not the melodic, aspect of sound sequences drove this effect. This indicates that newborns extract statistical regularities from structured contexts (real condition)—likely the rhythmic relationship between nonadjacent timing intervals—to predict upcoming events in the sequence. Conversely, musical expectations were reduced when regularities were weak or absent in random contexts (shuffled condition). As expected, local temporal information—the latency difference between two adjacent notes—was encoded while listening

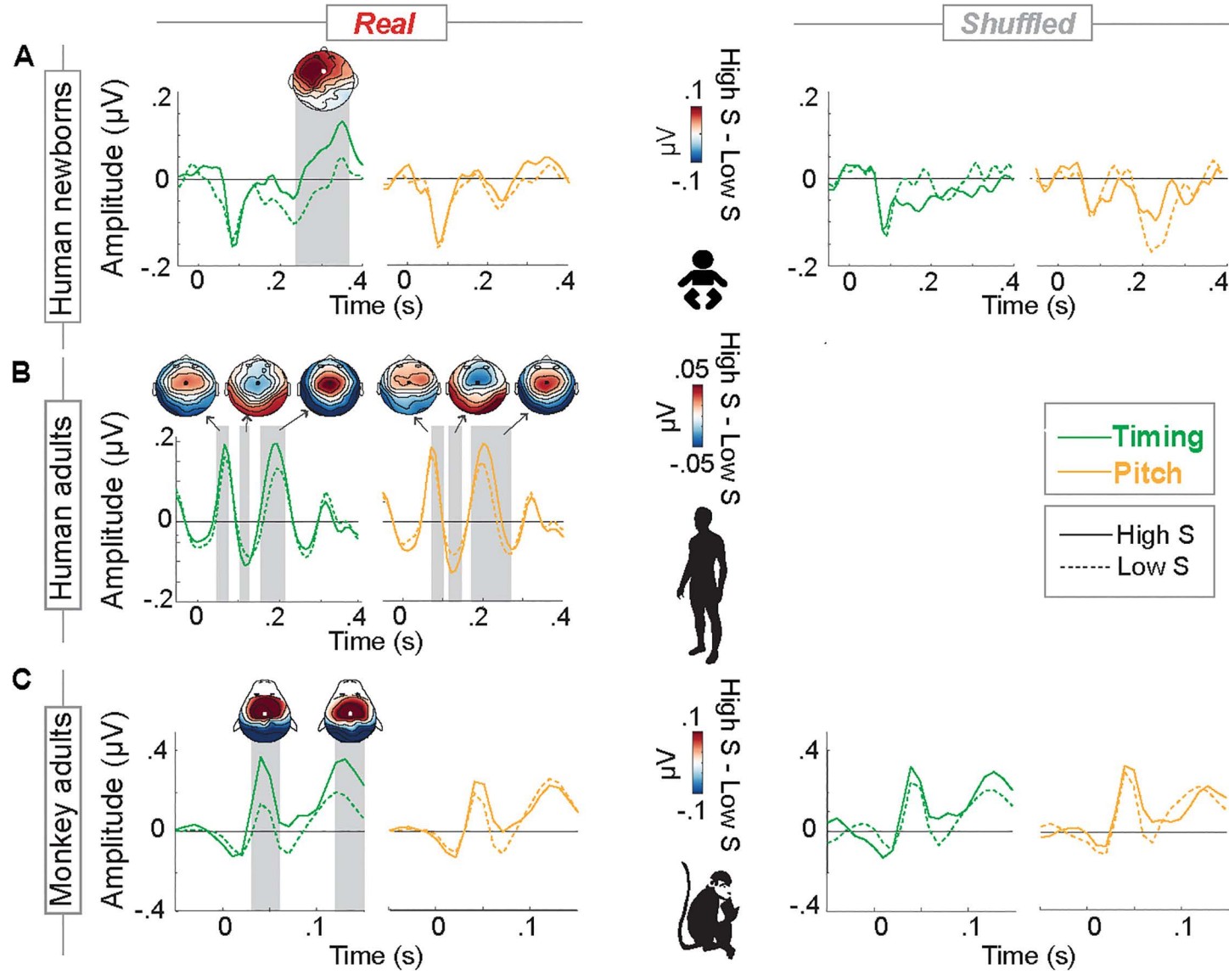

**Fig 3. Modulation of auditory event-related potentials as a function of surprise. (A)** Human newborns. Group-average (*n* = 49) ERPs (electrode Fz) evoked by notes carrying relatively high (continuous line) versus low (dashed line) St (green) and Sp (yellow) for real and shuffled music locked to the note onset (0 s). The amplitude of the P2 component was higher in response to notes carrying relatively high versus low temporal surprise (green) (from +.24 to +.37 s) for real (left) but not for shuffled music (right). No effect of pitch-related surprise was found. Gray windows highlight significant differences between low versus high surprise responses (cluster-corrected permutation tests over time across all electrodes). Topographies illustrate the amplitude difference between conditions in the time windows of identified clusters. See S3 and S4 Data. **(B)** Human adults. To assist the comparison with results of previous studies, we plotted group-average (*n* = 20) ERPs (electrode FCz) recorded from human adults listening to the same stimuli as the infants (reanalysis of data from [51]). Note that no shuffled stimuli were presented in this study. The amplitude of the P1-N1-P2 components was higher in response to notes associated with high than those with low temporal surprise (P1: from +.05 to +.07 s; N1: from +.11 to +.12 s; P2: from +.16 to +.21 s). Notably, human adults also exhibited sensitivity to pitch-related surprise, as indicated by enhanced P1-N1-P2 in response to notes carrying relatively high versus low surprise in pitch (Sp, yellow) (P1: from +.07 to +.09 s; N1: from +.12 to +.14 s; P2: from +.17 to +.27 s). **(C)** Adult Rhesus monkeys. Group-average (*n* = 2) ERPs (electrode FCz) recorded from Rhesus Monkeys listening to the same stimuli as the infants (reanalysis of data from [19]). The amplitude of the P1 and P2 components (P1: from +.03 to +.06 s; P2 from +.12 to +.15 s, frontal electrodes) was higher in response to notes associated with high than those with low temporal surprise for real, but not for shuffled music. Similarly to newborns, pitch-related surprise did not yield any significant modulation of the EEG amplitude.

to both real and shuffled music (see Fig 2C, IOI reduced model), independently of whether high-order structural patterns were present. These findings align with the idea that tracking event predictability relies on the ability to extract and represent structural information from the past context [62,63]. The reduced response in the shuffled condition reflects a down-weighting of such predictability-related response when the inferred stability, or precision, of the sensory input is low, and the present information does not conform with past experience [64,65].

This finding also brings novel evidence to our understanding of human rhythmic abilities present at birth. While rhythmic skills, such as sensitivity to isochrony and beat periodicity, are well-documented in infants at 5 months [66], at birth [43], and even in preterm infants [42], evidence regarding sensitivity to context-based probabilistic expectations remains elusive [46]. Here, we offer positive evidence. Using J. S. Bach's compositions with a variable range of IOIs, we show that newborns are not merely tracking isochrony and periodic patterns. They also process a higher-level feature, namely the probability of when the next event will occur based on a range of past different IOIs. This capacity in infants might build upon the well-documented sensitivity to isochrony and periodicity: in other words, an isochronous or periodic representation of a sequence might provide a temporal grid of predictable sound events, like a scaffold facilitating the segmentation and organization of more complex temporal and/or spectral patterns [67].

What could underlie such precocious rhythmic abilities? A potential candidate is the fetal sensory environment, which is characterized by the prominence of biological rhythms. This includes auditory stimulations (e.g., the mother's heartbeat [4]), as well as vestibular stimulations (e.g., associated with the regular pace of maternal gait [68–70]). An alternative possibility is that newborns developed such predictive skills through exposure to musical input during gestation [71]. This hypothesis, however, appears not to be supported by our supplementary analysis, demonstrating that estimating surprise values using a model pre-trained on a large musical corpus, reflecting prior exposure, produced results similar to those obtained from a model without pre-training (S4 Fig). Thus, fast statistical learning throughout the stimulus set provides a more parsimonious explanation. This is generally consistent with the existence of an inborn automatic statistical learning mechanism for sequence processing [72], and in line with recent EEG evidence of the neonates' ability to rapidly learn transition probabilities across different attributes of complex sounds, such as speech [73,74]. While we could not manipulate prenatal musical exposure, future research should systematically manipulate it under the hypothesis that greater musical exposure would lead to stronger neural encoding of musical expectations.

Turning to the functional significance of such precocious rhythmic abilities, we speculate that it might be key in the early development of cognition, not only as a precursor to higher-order statistical learning but also as a mechanism for orienting attention and organizing behavior in time [75]. In support of this idea, newborns can partially adapt spontaneous rhythmical behaviors, such as sucking, to external stimuli [76]; rhythmical rocking interventions on preterm infants improve orienting responses [77]; and vestibular rhythmical stimulation on preterm infants increases their adaptive breathing response, vital for organizing structured behaviors, such as feeding, early vocalizations, and interactions [78].

As opposed to rhythm, we found no evidence for neural encoding of local pitch intervals (IPI) or pitch-based probabilistic expectations (Sp) (note, however, that IPI neural tracking approached significance when compared to zero in the real condition). Whilst this is at first surprising based on past work [47–50], large variability and hardly detectable responses to pitch variations were also previously highlighted, suggesting that pitch neural tracking in newborns requires clear-cut pitch-change manipulations [61]. Such generally weak pitch encoding may stem from the fact that fetal hearing is heavily low-pass filtered in the womb [79], resulting in substantial attenuation of pitch details during gestation and slower maturation of pitch sensitivity. This is consistent with immature frequency-specific pathways and coarse frequency tuning at birth [80], as well as immature temporal resolution of different tones (see evidence from 6-month-old infants [81]). This factor, combined with the greater complexity of our stimuli compared with previous work, might explain the limited melodic tracking we observed. Indeed, our musical stimuli (across real and shuffled melodies) were characterized by sequences of multiple pitches ($N = 38$, mean height $= 73.04 \pm 6.32$; range: 55–93, in MIDI notation), IPIs ($N = 29$, mean interval size $= 4.66 \pm .4.12$; range: 0−28), and IOIs ($N = 30$, mean interval size $= .19 \pm .13$; range: .03–2.6) presented at varying tempi and rhythms. While

these features better approximate everyday music listening, they indeed also pose a greater computational challenge for the neonate's brain compared to traditional oddball paradigms using constant IOI and large infrequent pitch deviants. In sum, newborns' sensitivity to pitch appears far from sufficient for appreciating musical pitch regularities, which likely emerge through maturation and enculturation. According to this view, reports of musical memory from the womb to birth [82] are likely to rely primarily on timing rather than pitch information, a hypothesis that deserves further testing.

The dissociation observed between rhythmic and melodic statistical tracking might stem from their independent yet complementary neural implementations—respectively relying on temporal and content-based signaling along the auditory hierarchy [83–85]. This separation grants flexibility, allowing the brain to weight predictive signals by their reliability to optimize sequence perception [6]. Our findings suggest that the weighting of these two predictive processes is shaped by developmental refinement, with rudimentary pitch encoding at birth, eventually becoming as robust as temporal encoding later on. It is also possible that these two processes are differentially vulnerable to vigilance states. According to this hypothesis, during sleep, timing is favored over pitch because it is more salient and potentially linked to survival-relevant cues [75]. This would align with EEG studies on adults, suggesting that in-sleep perception and learning might be restricted to simple salient information [86] (see examples in music [87] and speech [88]). Future research should investigate whether melodic processing is modulated by sleep in newborns and whether it is similarly underweighted in sleeping adults. This would clarify whether this effect is truly lacking in newborns or is a consequence of how pitch-related information is processed during sleep.

From a phylogenetic perspective, the prominent perceptual role of rhythm observed in early phases of human ontogeny might piggyback on a more ancestral phylogenetically conserved sensitivity to rhythm (rather than melody) within the primate lineage. The ERP analysis showed greater amplitude of P1-P2 responses to temporally unexpected than expected notes but no modulation in the pitch dimension (Fig 3A). Given their similar broad frontal topography, these two peaks may reflect a single positivity with a similar underlying generator (as also discussed in [58,89]). They might also represent precursors to the adults' P1 and P2 components [90] (Fig 3B), possibly involving frontotemporal areas for sensory predictive processing and memory-based sequential integration [91–93]. Interestingly, the P1-P2 responses of both monkeys and human adults listening to the same stimuli presented to the newborns were also modulated by temporal surprise [19] (see the re-analyses in Fig 3B and 3C). Thus, the similarity in cortical responses to temporal but not pitch surprise across groups suggests rhythm as a primary perceptual cue in auditory sequence tracking. This does not imply that humans and monkeys generate rhythmic expectations through the same neural mechanism, even if the cortical responses are similarly modulated by temporally unexpected events. For instance, these responses might reflect the contribution of anticipatory mechanisms operating over different rhythmic features—ranging from periodicity and local temporal changes to statistical and hierarchical structures [94, 95]. Supplementary analyses suggest that, in newborns, probabilistic and local temporal information explain comparable amounts of EEG variance (Figs 2C and S3), whereas in monkeys, local temporal information contributes relatively more strongly (Figure S3 in [19]). Comparing distinct rhythmic computational models across phylogenetically close groups and as a function of exposure might shed light on the biological basis and evolutionary history of these different rhythmic capacities.

Regarding melodic expectations, the lack of significant melodic tracking observed in both human newborns and musically naïve monkeys (as opposed to human adults, Fig 3) leaves open the hypothesis that melodic sensitivity may not have emerged only in humans within the primate lineage but might potentially develop in other nonhuman primates given sufficient musical exposure. Testing this hypothesis across species could shed light on the role of experience in shaping the relative weighting of pitch- and timing-based expectations in auditory processing.

## Conclusions

Overall, this study provides neurophysiological evidence that tracking rhythmic statistical regularities is a capacity present at birth, whilst melodic tracking might not be, at least with respect to naturalistic musical stimuli, such as the ones

we used here. Future investigations should assess whether the observed dominance of rhythm over melody reflects state-dependent factors such as sleep or instead marks an early developmental bias that gradually shifts with experience toward the balanced sensitivity observed in adulthood.

## Methods

### Ethics statement

Written formal consent was obtained from the parent/guardian, and the infant's mother could opt to be present during the recording. The study fully complied with the World Medical Association Helsinki Declaration and all applicable national laws. Approval was granted by the Hungarian Medical Research Council, Committee of Scientific and Research Ethics (ETT TUKEB), ethics approval: IV/2199-4/2020/EKU.

### Participants

Sixty-four healthy full-term newborn infants (0–2 days of age, 30 male, and APGAR score 9/10) were tested at the Department of Obstetrics-Genecology, Szent Imre Hospital, Budapest. EEG data from 6 infants were corrupted, and 9 other infants did not complete the experiment. As a result, these data were not analyzed, leaving a total sample size of 49. The analyzed infants had a mean gestational age of 40 weeks (SD = 7.1 days) and a mean birthweight of 3468.1 g (SD = 398.2 g). All newborns had normal hearing and passed the Brainstem Evoked Response Audiometry (BERA) test.

### Stimuli

The stimuli consisted of 14 monophonic piano melodies used in [19] (details in S1 Table): 10 melodies (real music) composed by Johann Sebastian Bach (previously also used in [51]) and 4 control melodies (shuffled music) created by disrupting the pitch order and timing regularities of four of the original melodies (see below). The length of the melodies varied (average duration = 158.07 s ± 24.06), and the tempo ranged from 47 to 140 bpm (average tempo = 106.5 bpm ± 34.7). The four shuffled melodies were derived from four of the real melodies, specifically selected to represent those with the highest (melodies 05 and 08) and lowest (melodies 01 and 10) temporal-onset mean surprise. This selection was motivated by evidence suggesting that music with higher timing surprise elicits stronger brain responses in humans, aiming to balance these effects across both real and shuffled music. The shuffled melodies were matched to the real melodies in terms of pitch content, average note duration, and IOIs, but their structure was disrupted in two key musical dimensions. Pitch regularities were altered by reordering the temporal sequence of the original notes. Rhythmic patterns were disrupted by creating a new set of IOIs drawn from a Gaussian distribution centered around the original mean IOI, with an added variation based on the difference between the mean and the minimum IOI. These randomly generated IOIs were then adjusted in MuseScore software (version 3.3.4.24412, https://musescore.org) to align with 16th-note quantization, preserving integer ratios. In MuseScore, the MIDI velocity (which correlates to note loudness) was standardized to a constant value of 100, and piano sound waveforms were synthesized with a 44,100 Hz sampling rate. Each melody was preceded and followed by a beep (800 Hz pure tone, linearly ramped with a 5 ms fade-in and fade-out) and a 5-s silence, following the structure: beep-silence-music-silence-beep. The resulting audio files were converted to mono and amplitude-normalized by dividing by the standard deviation using Matlab (R2019, The MathWorks, Natick, MA, USA).

### Information dynamics of music model

Stimuli were analyzed using the IDyOM model (https://www.marcus-pearce.com/idyom/), which predicts note-by-note unexpectedness (surprise) and uncertainty (entropy). IDyOM is a variable-order Markov model that learns statistical patterns from musical sequences. It generates probability distributions for each new note based on prior context and outputs surprise (S) and entropy (E) over time. Surprise measures the unexpectedness of an event at time 't' once it has

occurred. Entropy reflects the uncertainty about the event at 't' before it occurs based on the probability distribution of all potential notes considering all observations prior to the event at 't'. The model incorporates both short-term and long-term contexts, with the short-term model trained on the current sequence and the long-term model on prior musical exposure. To simulate the statistical knowledge that the newborns would acquire through mere exposure to the stimuli, predictions were derived from a combination of short and long-term models, with the latter being trained only on the stimuli used in the experiment, i.e., via resampling (10-fold cross-validation) (in IDyOM terminology: no pretraining, "both+" model configuration). IDyOM can account for many aspects of music, but here, we focused on two key dimensions that best describe piano monophonic melodies: pitch and timing. To this end, time series representing pitch and inter-onset interval ratios (using separate 'cpitch' and 'ioi-ratio' IDyOM viewpoints) were analyzed independently by IDyOM to calculate note-by-note surprise (S) and entropy (E) for both pitch ($S_p$, $E_p$) and timing ($S_t$, $E_t$). These were then combined to determine the joint (sum) probability for each note (S, E). To simulate the long-term statistical knowledge of music possibly acquired by infants in the womb, we run control analyses by using S and E estimates derived by an IDyOM model pre-trained on a large corpus of music (comprising 152 Canadian folk songs, 566 German folk songs from the Essen folk song collection, and 185 J. S. Bach chorale melodies (as in previous applications [19,51,96]).

## Procedure

As a common procedure in EEG studies in newborns, infants were asleep during the EEG recording and stimulus presentation. Stimuli were presented using a Maya 22 USB external soundcard and ER-2 Insert Earphones (Etymotic Research, Elk Grove Village, IL, USA) placed into the infants' ears via ER-2 Foam Infant Ear-tips. The melodies were presented at a comfortable intensity (about 70 dB SPL). Two sets of the 14 melodies were presented in a randomized order within each set, ensuring that each infant listened to each melody at least once, with some melodies being heard twice. None of the infants heard the full set of 14 melodies twice (range 14–25), and on average, they had 1.18 (SD = 1.4) repetitions. The presentation was implemented in Matlab (R2014, The MathWorks, Natick, MA, USA) and Psychtoolbox (version 3.0.14). EEG was recorded throughout the stimulus presentation. The inter-stimulus interval between melodies (ISI, offset to onset) was 900–1,300 ms (random with even distribution, 1 ms step). The experiment took 45 min overall, including both preparation and stimulation.

## Data recording and preprocessing

An ActiChamp Plus amplifier with a 64-channel sponge-based electrode system (saltwater sponges and passive Ag/AgCl electrodes, R-Net) and a Brain-Vision Recorder were employed to record EEG (Brain Products GmbH, Gilching, Germany). The sampling rate was 500 Hz with a 100 Hz online low-pass filter applied. Electrodes were placed according to the International 10/10 system. The Cz channel served as the reference electrode while the ground electrode was placed on the midline of the forehead. During the recording, impedances were kept below 50 kΩ.

Data were preprocessed and analyzed in MATLAB R2019. For the analysis, we applied a fully data-driven pipeline for preprocessing EEG data, combining open-access denoising algorithms, similar to previous studies dealing with noisy EEG recordings [19,59]. The analysis used Fieldtrip [97] and EEGLAB toolboxes (http://sccn.ucsd.edu/). The continuous EEG data were bandpass filtered between 1 and 30 Hz (Butterworth filter, zero-phase, order 3), down-sampled to 100 Hz, and segmented into epochs from the onset to the offset of each melody, separately. Before re-referencing the data to the average of a set of electrodes ('F9', 'F10', 'P9', 'P10', and 'Iz'), faulty or noisy electrodes were temporarily discarded to prevent noise contamination across electrodes. Specifically, for each electrode, the mean, standard deviation, and peak-to-peak values were calculated across time within each trial. If any of these values deviated by more than 2.75 standard deviations from the mean of other electrodes, the electrode was flagged as noisy/faulty. This process was repeated until a distribution without outliers was obtained. The data were then further denoised in EEGLAB using the Artefact Subspace Reconstruction (ASR) algorithm [98] (threshold value 5 previously validated for both adult human and monkey EEG data

[19]). Eye-movement artifacts were corrected using the ICLabel algorithm in EEGLAB. After performing independent component analysis (ICA) with EEGLAB's 'runica' function, independent components labeled by ICLabel as 'eye movements' (with > 90% likelihood) were rejected. Subsequently, electrodes that were initially excluded (due to being faulty or noisy) were interpolated by replacing their voltage with the average voltage of the (preprocessed) neighboring electrodes (18 mm distance, including 8 electrodes on average). If, following the above preprocessing, noisy electrodes were still automatically identified, the interpolation step was repeated (the number of such iterations varied between 1 and 2).

## TRF analysis

We employed Temporal Response Functions (TRF) to model EEG responses to the continuous acoustic and musical features of the presented stimuli using the mTRF MATLAB toolbox [53]. Each stimulus feature (as listed below) was normalized across time for each melody, ensuring that the root mean square of each feature was 1. A forward model was run to predict the ongoing EEG response from the stimulus features, with a time lag window of −50 to +400 ms to capture EEG fluctuations related to changes in the stimulus. This time window was sufficiently large to encapsulate well-known ERP-like modulations of EEG signals that are known to drive the variance modeled by TRF. Ridge regression was used to prevent overfitting (lambda range: $10^{-4}$ to $10^{8}$). TRFs were fitted to all melodies (pooled real and shuffled melodies) using leave-one-melody-out cross-validation, and the EEG time course of the left-out melody was predicted. Note that the correlation values are typically calculated between EEG signals and their predictions by considering single-participant EEG signals, which might carry much noise (especially if recorded from newborns). As such, EEG prediction correlations are variable between participants largely due to the variable SNR of the EEG signal across participants (as every prediction is correlated with a different EEG signal). To overcome this issue, we averaged all participants' EEG timeseries data to form a single EEG 'super-subject' data timeseries, which we refer to as 'ground-truth EEG'. Then, per each participant, melody, and electrode, prediction accuracy was quantified by calculating Pearson's correlation between the predicted and ground-truth EEG data.

We tested the contribution of high-level probabilistic musical expectations to the predicted EEG in addition to that of the low-level acoustic features in both the timing and pitch dimensions. Feature selection was based on the approach used in [19]. We additionally tested for the contribution of local changes in timing and pitch, such as IOI and IPI (measured in ms and absolute number of semitones, respectively), as these features are to some extent correlated with surprise values in naturalistic music (e.g., relatively larger temporal or pitch deviations tend to be relatively more unexpected, particularly in structured compositions like Bach's) [99].

Thus, we run a full model including low-level acoustic features (acoustic onset, spectral flux, as well as IOI and IPI) and high-level probabilistic musical features with impulses at the note onsets but whose amplitudes are set to the pitch and onset surprise and entropy values from IDyOM (surprise pitch, surprise timing and entropy pitch, entropy timing—Sp, St and Ep, Et). A control analysis, adding envelope and its half-rectified derivative as part of low-level acoustic regressors, yielded a pattern of results similar to the main analysis (see S5 Fig).

Although, as shown in Fig 1C, the correlations across regressors in our stimuli are small (<0.3) to moderate (<0.5), we used a variance partitioning approach that accounts for shared variance across regressors, allowing us to estimate their unique contributions to neural responses. We acknowledge, however, that a more direct way to assess the unique contribution of individual features is through causal manipulation, as demonstrated by the "model-matched" stimulus approach [100]. Thus, to assess the unique neural encoding of a single or a set of stimulus features, we subtracted the prediction accuracy of several reduced models from the full model (containing all features). We then analyzed the Δr values obtained for each reduced model. Note that we used the term 'reduced' to indicate a model in which a given predictor (or a set of predictors) is temporally shuffled to estimate its unique contribution to the full model. The reduced models had the same dimensionality as the full model, but the feature/s of interest was/were randomized in time whilst preserving the onset times. We tested five reduced models: 1) A probabilistic music model where the high-level features (St and Et and Sp

and Ep) were randomized to assess the overall effect of adding surprise and entropy estimates to low-level features to the neural tracking; 2) A probabilistic timing model with randomized St and Et; 3) A probabilistic pitch model with randomized Sp and Ep; 4) A local timing model with randomized IOI; and 5) A local pitch model with randomized IPI. To compare across the different models, for each participant, condition, and TRF model, Δr values were averaged across 25% of channels with the highest prediction accuracy in the full model and across the real and the shuffled conditions. These values were then entered into linear mixed-effects regressions. Note that ROIs were defined as the top 25% of electrodes showing the highest correlation values in the full model, averaged across conditions, ensuring independence from both condition (real versus shuffled) and regressor type (surprise pitch versus surprise timing). Using alternative thresholds (top 10% or 50%) yielded the same pattern of results, confirming that findings were robust to ROI definition.

## Statistical analysis

Statistical analyses were run in R (version 4.1.3, 2022-03-10) and included nonparametric tests or linear mixed-effects models (lme4 package). All models included Random Effects of Infants and Melodies (IDs 1–14). The Fixed effects included Condition (real/shuffled) and TRF model (depending on the comparison; see Results). Statistical significance was evaluated by likelihood-ratio tests ($\chi^2$) conducted using the 'anova' function (stats package). Follow-up contrasts were conducted using the 'emmeans' package and the Tukey method to account for the increased risk of type I error resulting from multiple comparisons. Adjusted $p$-values were calculated to determine significant differences between conditions. A significance level of a = 0.05 was used. All linear mixed-effects models (LMMs) report fixed-effect estimates (b) along with their standard errors (SE) and $t$-values, with degrees of freedom estimated via Satterthwaite approximation when applicable.

When a direct test of differences was needed, the nonparametric Wilcoxon signed-rank test was used. For these, results are reported as W-values, indicating the sum of ranks of signed differences.

## ERP analysis

Event-related potential (ERP) analyses were performed by segmenting the EEG data into 600 ms epochs, beginning 100 ms before the onset of each note and ending 500 ms after the onset. Epochs were baseline corrected using a 50 ms window before the note onsets, and trials that deviated from the mean by more than 2.5 the average standard deviation were rejected (3.45 ± 1.12% of the trials per subject). To assess ERP modulation based on note surprise, we selected the notes with the highest and lowest 20% surprise (high S and low S) values, separately for each melody, as assessed by the IDyOM. For each subject, epochs were trimmed to a window of −50 + 400 ms relative to note onset and averaged by high/low S condition, separately for real and shuffled melodies. Cluster-based permutation testing [101] was used to account for multiple comparisons across adjacent time points and electrodes. Clusters of adjacent timepoints and neighboring electrodes (at least three) associated with significant ($p$-values < 0.025) differences across conditions were formed. A cluster-level threshold of $p < 0.05$ was applied to the $t$-statistic, and the Monte Carlo method (1,000 iterations) was used to estimate the null distribution of this statistic. To assist comparability with the previous work, we re-analyzed the EEG data recorded from human [51] and monkey [19] adults following the same pipeline described here (both datasets are open source). Note that for the monkey data, as in the original work, clusters were identified separately for each animal (across 22 sessions) and considered significant only when exhibited by both animals (conjunction analysis).

## Supporting information

**S1 Fig. Summary statistics (mean and variance of the amplitude envelope across frequency bands) of stimuli.**
To extract the envelope associated with each frequency band, we bandpass-filtered the musical stimuli into 128 logarithmically spaced frequency bands ranging from 100 to 8,000 Hz using a gammatone filter bank. We then computed the

amplitude envelope of each band as the absolute value of the Hilbert-transformed signal over time. The envelope mean (left) and variance (right) are shown as a function of frequency band. Thin lines represent individual melodies (real and shuffled), while thick lines indicate the average across melodies for Real (solid line) and shuffled (dotted line) conditions. Note that the real and shuffled conditions show comparable envelope means and variances. See S5 Data.
(TIF)

**S2 Fig. Difference in prediction accuracy (Δr) between real and shuffled conditions across participants and electrodes for each reduced model.** The 2D matrices display electrodes on the Y-axis and participants on the X-axis, with colors coding the difference in Δr values (full—reduced model) between real and shuffled conditions. Note that positive values (red color coded) indicate a greater contribution of the real compared to the shuffled condition. Near-zero values (white color coded) indicate similar contributions across conditions. **(A)** Unique contribution of high-level musical features. Between condition difference in Δr (full—reduced model assessing the unique contribution of high-level musical features—St, Et, Sp, and Ep). To facilitate visualization, we display the labels of 16 representative electrodes (out of 63) on the $y$-axis, along with their corresponding position on the EEG cap (bottom). **(B)** Unique contribution of timing and pitch-related features. Between condition difference in Δr (full and reduced models separately assessing the unique contribution of St and Et, Sp and Ep, IOI, and IPI). See S6 Data.
(TIF)

**S3 Fig. (A) Length of preceding and subsequent IOIs as a function of rhythmic surprise and condition.** Notes carrying high surprise are often preceded by relatively larger IOIs. A linear mixed model predicting the preceding IOI, with factors condition (real/shuffled) and surprise level (high/low), yielded no main effect of condition ($\chi^2(1) = 1.12$, $p = 0.29$), but a significant main effect of surprise level ($\chi2(1) = 16.89$, $p < .001$), and an interaction of condition and surprise level ($\chi2(1) = 18.12$, $p < .001$). This indicates that larger IOIs generally anticipate notes carrying high surprise, more so in real than in shuffled music (left panel). Conversely, the same analysis predicting the subsequent (rather than preceding) IOI, yielded a nearly significant effect of surprise level ($\chi^2(1) = 3.52$, $p = 0.06$) but no main effect of condition ($\chi^2(1) = 0.008$, $p = 0.90$) and no interaction ($\chi^2(1) = 0.70$, $p = 0.40$). This indicates that notes carrying high surprise tend to be followed by larger IOIs, but comparably across real and shuffled music (right panel). See S7 Data. **(B)** Unique contribution of St and Et is independent of preceding or subsequent IOI. To distinguish neural tracking of rhythm from spurious modulations of event-related potentials (ERPs) attributable to overlapping (i.e., temporally proximal) neural responses, we re-run the main analysis, adding the length of the subsequent IOI as a regressor in the mTRF. We thus run a full model with the following regressors: onset, spectral flux, inter-pitch interval, preceding IOI, subsequent IOI, Sp, Ep, St, and Et. We then computed three reduced models, each randomizing one of the following regressors: 1) St and Et, 2) preceding IOI, and 3) subsequent IOI. The results of this control analysis confirm a unique contribution of St and Et features to the neural response beyond the contribution of subsequent and preceding IOI. The plot shows the topographical maps representing group-average Δr resulting from the difference between the full and the three reduced models across real and shuffled conditions.
(TIF)

**S4 Fig. Control analyses.** No effects of IDyOM statistical knowledge on EEG prediction accuracy. We compared the effect of deriving surprise estimates by training IDyOM on either the experimental stimuli alone (left panel) or on the experimental stimuli, as well as an additional corpus of Western tonal music (right panel). For each panel, we plot the difference in EEG prediction accuracy (Δr) between the full and the reduced models (randomizing St, Et, Sp, and Ep). Dots represent the grand-average mean Δr computed across all channels and melodies (left top panel, with associated topographical maps) for real (red) and shuffled (gray) music. Error bars represent bootstrapped 95% CI. The absence of differences in predicting neural responses between pre-trained and nonpre-trained model configurations suggests that incorporating pretraining to estimate surprise and entropy values does not enhance the prediction of EEG data. This may be due to the high correlation between the estimates derived from the two IDyOM configurations, leading to similar EEG predictive

power. Additionally, it may indicate that Bach's music contains sufficient rules and statistical regularities, allowing the model to learn these directly from the stimulus set, rendering pretraining on the large music corpus redundant to predict brain signals. See S8 Data.
(TIF)

**S5 Fig. Control analysis.** Replication of the results reported in Fig 2C, here adding envelope and its half-wave rectified derivative as predictors in the full model. We repeated the analysis using an enriched acoustic model, thus adding envelope and its half-wave rectified derivative to the already used acoustic regressors (onsets, spectral flux, ITI, and IOI). This additional analysis confirms the robustness of our results: the main findings remain overall unchanged, indicating a unique contribution of the high-level music regressors in the real but not the shuffled music condition, specifically driven by the St and Et regressors. See S9 Data.
(TIF)

**S1 Table. Characteristics of the experimental stimuli.**
(DOCX)

**S1 Data. Excel file containing the numerical data values for Fig 1B and 1C.** Sheet1: average surprise and entropy values of each note, computed separately for pitch and timing, within each melody. Sheet2: values used for Pearson's correlation between the stimulus features of all melodies: inter-pitch-interval (IPI), inter-onset-interval (IOI), and surprise and entropy associated with timing (S t and E t) and pitch (Sp and Ep).
(XLSX)

**S2 Data. Excel file containing the numerical data values for Fig 2B, 2C, and 2D.** Sheet1: EEG prediction accuracy (*r* values) for each infant and melody of the full model, as well as the difference between the full and the reduced model (Δr values), assessing the unique contribution of high-level musical features (St, Et, Sp, and Ep). Sheet 2: EEG prediction accuracy (*r* values) for each infant and melody of the full model, as well as the difference between the full and the reduced model (Δr values), assessing the unique contribution of timing (St and Et) and pitch-related (Sp and Ep) high-level features. Sheet 3: EEG prediction accuracy (*r* values) for each infant and melody of the full model, as well as the difference between the full and the reduced model (Δr values), assessing the unique contribution of timing (IOI) and pitch-related (IPI) low-level features.
(XLSX)

**S3 Data. MATLAB file containing the numerical values underlying Fig 3A, representing ERPs evoked by notes with relatively high versus low *pitch* surprise (Sp) for real and shuffled music, time-locked to note onset.** The data are stored as a 4D matrix with dimensions subject × condition × channel × time, where conditions 1–4 correspond to Low Sp (real), High Sp (real), Low Sp (shuffled), and High Sp (shuffled). Data can be opened using nonproprietary software such as R or Python.
(MAT)

**S4 Data. MATLAB file containing the numerical values underlying Fig 3A, representing ERPs evoked by notes with relatively high versus low *timing* surprise (St) for real and shuffled music, time-locked to note onset.** The data are stored as a 4D matrix with dimensions subject × condition × channel × time, where conditions 1–4 correspond to Low St (real), High St (real), Low St (shuffled), and High St (shuffled). Data can be opened using nonproprietary software such as R or Python.
(MAT)

**S5 Data. Excel file containing the summary statistics (mean and variance of the amplitude envelope across frequency bands) of stimuli shown in S1 Fig.**
(XLSX)

                                                    

**S6 Data. Excel file containing the numerical values underlying S2 Fig.** The Δr values (full—reduced model) for each of the 5 plots, corresponding to 5 different reduced models, are stored in 5 different spreadsheets containing values for each subject × condition (real/shuffled) × electrode.
(XLSX)

**S7 Data. Excel file containing the numerical data values for S3A Fig.** Surprise associated with timing, length of preceding IOI, as well as length of the subsequent IOIs for each note of all melodies).
(XLSX)

**S8 Data. Excel file containing the numerical data values for S4 Fig.** Difference of EEG prediction accuracy for each infant between the full and the reduced model (Δr values), assessing the unique contribution of high-level musical features (St, Et, Sp, and Ep) obtained from an enculturated IDyOM model trained on an extra music corpus.
(XLSX)

**S9 Data. Excel file containing the numerical data values for S5 Fig, with the full model additionally containing envelope and its derivative.** Sheet1: EEG prediction accuracy (r values) for each infant and melody of the full model, as well as the difference between the full and the reduced model (Δr values), assessing the unique contribution of high-level musical features (St, Et, Sp, and Ep). Sheet 2: EEG prediction accuracy (r values) for each infant and melody of the full model, as well as the difference between the full and the reduced model (Δr values), assessing the unique contribution of timing (St and Et) and pitch-related (Sp and Ep) high-level features. Sheet 3: EEG prediction accuracy (r values) for each infant and melody of the full model, as well as the difference between the full and the reduced model (Δr values), assessing the unique contribution of timing (IOI) and pitch-related (IPI) low-level features.
(XLSX)

## Acknowledgments

We thank the Brain and Machines Flagship Programme of the Italian Institute of Technology (https://www.iit.it/our-research) for the support.

## Author contributions

**Conceptualization:** Roberta Bianco, Giacomo Novembre.

**Data curation:** Roberta Bianco, István Sziller, Gábor P. Háden.

**Formal analysis:** Roberta Bianco.

**Funding acquisition:** Roberta Bianco, Brigitta Tóth, István Winkler, Giacomo Novembre.

**Investigation:** Roberta Bianco, Brigitta Tóth, István Sziller, Gábor P. Háden.

**Methodology:** Roberta Bianco, Felix Bigand, Trinh Nguyen, Giacomo Novembre.

**Project administration:** Roberta Bianco, Brigitta Tóth, Giacomo Novembre.

**Resources:** Brigitta Tóth.

**Software:** Roberta Bianco.

**Supervision:** Giacomo Novembre.

**Validation:** Roberta Bianco, Felix Bigand.

**Visualization:** Roberta Bianco.

**Writing – original draft:** Roberta Bianco, Giacomo Novembre.

**Writing – review & editing:** Roberta Bianco, Brigitta Tóth, Felix Bigand, Trinh Nguyen, Gábor P. Háden, István Winkler, Giacomo Novembre.

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
