## [Editor Report · Decision Letter 0]

11 Mar 2025

Dear Dr Bianco,

Thank you for submitting your manuscript entitled "Human newborns form musical predictions based on rhythmic but not melodic structure" for consideration as a Research Article by PLOS Biology.

Your manuscript has now been evaluated by the PLOS Biology editorial staff as well as by an academic editor with relevant expertise and I am writing to let you know that we would like to send your submission out for external peer review.

Once your full submission is complete, your paper will undergo a series of checks in preparation for peer review. After your manuscript has passed the checks it will be sent out for review. To provide the metadata for your submission, please Login to Editorial Manager (https://www.editorialmanager.com/pbiology) within two working days, i.e. by Mar 13 2025 11:59PM.

Kind regards,

Christian

Christian Schnell, PhD

Senior Editor

PLOS Biology

cschnell@plos.org

---

## [Decision Letter · Decision Letter 1]

24 Apr 2025

Dear Dr Bianco,

Thank you for your patience while your manuscript "Human newborns form musical predictions based on rhythmic but not melodic structure" was peer-reviewed at PLOS Biology. It has now been evaluated by the PLOS Biology editors, an Academic Editor with relevant expertise, and by several independent reviewers.

In light of the reviews, which you will find at the end of this email, we would like to invite you to revise the work to thoroughly address the reviewers' reports.

As you will see below, the reviewers think that the study is overall well executed and provides important insights. The main concerns which the reviewers share concern the interpretation of the results (for example about the comparisons between species and to human adults), but also the model.

Given the extent of revision needed, we cannot make a decision about publication until we have seen the revised manuscript and your response to the reviewers' comments. Your revised manuscript is likely to be sent for further evaluation by all or a subset of the reviewers.

**IMPORTANT - SUBMITTING YOUR REVISION**

*Re-submission Checklist*

*Published Peer Review*

*PLOS Data Policy*

*Blot and Gel Data Policy*

Sincerely,

Christian

Christian Schnell, PhD

Senior Editor

PLOS Biology

cschnell@plos.org

REVIEWS:

Reviewer #1: This manuscript addresses an important and timely question in developmental neuroscience and music cognition: whether the human ability to anticipate musical structure is already present at birth. The authors use temporal response functions (TRFs) to probe rhythmic and melodic encoding in newborns exposed to real versus temporally shuffled musical stimuli.

The study is built on a particularly rich and valuable dataset — EEG recordings from 49 newborns, an impressive achievement in itself. The main finding — that newborns encode probabilistic rhythmic regularities in real music, but not melodic ones — is highly interesting. It provides compelling evidence that rhythm-based statistical learning is already functional at birth.

We have some concerns regarding the channel selection procedure, which should be reworked.

Moreover, the interpretative sections of the manuscript, especially the conclusion, do not fully do justice to the depth of the results. The discussion often gets sidetracked, particularly by extended considerations of sleep state, while the core implications — the precedence of rhythmic over melodic processing — are underdeveloped. Several rich interpretive avenues (e.g., prenatal experience, cognitive cost of temporal vs. spectral prediction, the nature of the "structured context") are left largely unexplored.

Finally, we are also unconvinced by the relevance of the comparison with non-human primates in this context. While it supports the analytical framework, it does not substantially advance a phylogenetic interpretation — which, in any case, does not seem to be the central focus of the study.

Major Comments

1. Channel selection procedure:

While it is understandable that neonatal EEG signals are inherently noisy, the method used for channel selection—choosing the top 25% based on correlation—appears suboptimal. In particular, it introduces a form of double-dipping in the context of the linear mixed-effects model applied to delta-r (> 0). Indeed, currently the channels are selected based on the highest r-values in the full model, thus the following statistical test comparing full vs. reduced models is not valid. A cluster-based permutation-based approach could offer a more statistically grounded way to assess whether models including high-level features outperform controls. Alternatively, permuting the regressors of interest 1000 times (for instance) would allow identification of channels that respond significantly, rather than relying on an arbitrary percentile threshold.

2. Data visualization in supplementary material:

It would also be valuable to include, as supplementary material, a 3D visualization of the delta-r values—participants on the Y-axis, channels on the X-axis, and delta-r represented by color—for both conditions (music and shuffled) and both models (high-level, low-level). This would allow for a direct inspection of the TRF results without averaging over channels or participants, helping assess the robustness of the effects.

3. Acoustic model improvements:

The low-level acoustic model should be refined. Including at minimum the amplitude envelope and its half-wave rectified derivative would align better with known auditory encoding mechanisms and better capture music-related acoustic dynamics. These features have been shown to strongly modulate auditory responses in both music and speech and may coincide with tempo information in the stimuli. See, e.g., Di Liberto et al., eLife 2020 or Robert et al., bioRxiv 2024 (see below for references).

4. Rewriting the conclusion:

The current conclusion includes some redundancy and underplays the main theoretical contributions. Rather than focusing so heavily on the impact of sleep state, the authors should more directly address:

o Why rhythm may be more predictable than melody (e.g., prenatal exposure to rhythmic patterns through maternal physiology; possible cognitive prerequisites for pitch processing…). See e.g. Dehaene et al., 2015, Neuron for leads on this topic.

o What exactly distinguishes the "real" from the "shuffled" music, beyond surprise values? Could the authors provide concrete descriptions or statistical measures of this structured context? See, e.g., the work of McDermott.

o Whether newborns' predictions are based on prior exposure (e.g., in utero learning) or fast learning during the experiment. If the IDyOM model is trained on the same material as the infants hear, this distinction is essential and warrants more discussion, possibly including references on short context auditory learning in newborns.

o The study would benefit from a clearer contrast with previous work, particularly regarding the shift from detecting isolated deviants to modeling continuous, event-by-event predictions, and from using simple, sparse stimuli to rich, fast musical input — which may challenge newborns' cognitive processing capacity.

Minor Comments

1. In the introduction, the comparison between newborns and monkeys may confuse the notion of "innate" capacities, particularly as fetuses are already exposed to music and speech. A brief clarification on what "innate" is taken to mean in this context would help.

2. Relatedly, drawing on developmental literature about early temporal processing might provide a more relevant theoretical framework than uniquely the primate comparison.

3. Could the authors provide more interpretation of the correlations between stimulus features? It would help understand to what extent they are separable in the model.

4. Consider comparing model performance (correlation values) with prior EEG TRF studies (e.g., Di Liberto et al., 2020, eLife; Robert et al., 2024, bioRxiv) to contextualize what constitutes a "reasonable" model accuracy.

5. The term "reduced model" is potentially misleading, as its size is not reduced but its meaningful content is randomized. Consider rewording to avoid confusion.

6. Figure 1D lacks the representation of the full model, making it unclear for readers.

7. Typo in Figure 2B (x10^-3 missing on the right axis).

8. Typo in Figure 2A: label should read "IPI" instead of "ITI".

9. Regarding ERPs: are unexpected pitch and unexpected timing correlated? A short clarification would be helpful.

10. Does musical tempo affect time prediction accuracy? If so, could the authors explore whether infants show a preferred tempo for prediction?

11. Procedure: why are some melodies heard twice? Could this repetition facilitate learning and influence the results?

References:

Di Liberto, G. M., Pelofi, C., Bianco, R., Patel, P., Mehta, A. D., Herrero, J. L., De Cheveigné, A., Shamma, S., & Mesgarani, N. (2020). Cortical encoding of melodic expectations in human temporal cortex. eLife, 9, e51784. https://doi.org/10.7554/eLife.51784

Robert, P., Van Cang, M. P., Mercier, M., Trébuchon, A., Bartolomei, F., Arnal, L. H., ... & Doelling, K. (2024). Multi-stream predictions in human auditory cortex during natural music listening. bioRxiv, 2024-11.

Dehaene, S., Meyniel, F., Wacongne, C., Wang, L., & Pallier, C. (2015). The Neural Representation of Sequences: From Transition Probabilities to Algebraic Patterns and Linguistic Trees. Neuron, 88(1), 2-19. https://doi.org/10.1016/j.neuron.2015.09.019

Reviewer #2: This paper presents results from a novel EEG study with sleeping newborn humans. The babies were presented with normal and shuffled experts of monophonic piano versions of Bach melodies. The authors consider both low level features, like inter-pitch-intervals and inter-onset-intervals as well as "higher level" quantifications of entropy and surprisal (generated from IDyOM) when testing models for predicting infant neural responses using mTRFs. Results suggest that EEG data were accuracy predicted using models that included both low and high level features. High level features added significantly to prediction accuracy but only for real and not shuffled melodies. This effect seems especially driven by high-level timing features but perhaps not high-level pitch features. ERP analyses tell a similar story - notes that are higher in temporal surprise generate a larger P1/P2-ish response than notes low in temporal surprise for real but not shuffled sequences (no such effects with pitch surprisal). The authors compare these findings to re-analyzed data from awake adults (hearing the "real" stimuli) and awake rhesus monkeys (hearing the same "real" and "shuffled" stimuli).

Overall, this is a very interesting and worthwhile study that answers some questions about early rhythm perception, raises some new interesting questions about how predictions are made about pitch vs timing in auditory stimuli, and uses compelling and cutting edge analysis techniques to explore these questions. I particularly like the use of real vs shuffled music. The authors present an interesting and thoughtful discussion.

In that discussion, they highlight something I was thinking about as I read the paper - that their comparison to adult and monkey data should be taken with a grain of salt given that the infants were the only group out of the three that were asleep. I agree that this is really important to highlight and appreciate the discussion provided. Is the "unimportance" of pitch here the result of development/experience or related to attentive/inattentive processing? I have to say, if the authors had the means to collect a sample of sleeping adults hearing both the shuffled and real stimuli, that would greatly strengthen the impact of this paper. I also realize that this was already a ton of work and is enough "as is", so if the authors choose to suggest running sleeping adults through this paradigm as a future direction (which they do) they may need to tone down some of their conclusions - for example, the entire final paragraph of the discussion.

In the introduction, it would be helpful to spend a bit more time explaining probabilistic regularities of melodies. I'm having a hard time understanding how we could possibly make these predictions about pitch without exposure to a musical system. Is the point that we see lower surprisal for pitch if the same kinds of intervals happen again and again within a sequence? This must take at least a bit of time to establish. It would be helpful to clarify this for those who may be unfamiliar with IDyOM/music theory, and it would be very helpful to link readers to your specific set of stimuli (that link to the Bach website was quite a rabbit hole). There was some discussion in the paper about how the IDyOM model using only this small set of stimuli came up with similar numbers compared to a model using a large corpus. But even still, the first time infants hear a few of these melodies, they can't be expected to already have accrued enough information about pitch regularities to make useful predictions. Melodic pitch information is mostly filtered out in the womb, especially more so than timing information (something that could be discussed), so even papers showing musical memory from womb to birth likely rely on timing information. Anyway, all that to say, did you consider exploring whether infants begin to rely more on higher-level pitch patterns in the last few trials than in the first few trials?

Real and shuffled stimuli differ broadly in entropy and surprisal. Is there any argument to be made for exploring how entropy and surprisal (regardless of shuffled/real categorization or within shuffled/real categories) relates to mTRF prediction accuracy? What is the value of considering surprisal and entropy separately in this paper? Do they tell us different stories? If not, why not pick one? If yes, the difference and value of presenting both is unclear.

These results raise interesting questions about individual differences - thank you for plotting individual data on the figures. Is it worthwhile exploring whether the babies who rely on higher level features the most for timing do the same for pitch? You mention that gestational age may explain some variability here. Do you have that information? Can you test that prediction? It might also be worth citing François et al., 2017, who report that newborn infant EEG responses to continuous song predict language outcomes in toddlerhood (DOI:10.1038/s41598-017-12798-2).

Minor comments:

- Can you clarify with the ERP analyses if the results are on Fz, as shown in the figure? Why was this electrode selected? It is being compared to Fcz in adults and monkeys. Is there an argument for presenting the same electrode in these figures across groups?

- It might be nice to add more discussion about primate rhythm perception/production in the intro. I was more familiar with behavioural work showing that primates are not good at predictively tapping, but learned a bit from this paper about newer work showing that beat perception may still be happening in some primates. Elaborating on how beat perception/productions in humans compared to non-human primates in the intro would be helpful given the phylogenetic arguments in the discussion

- This paper appears to argue that the predictive power of the IDyOM outputs suggests that infants are attuned to statistical regularities in musical beats. These statements about statistical learning have me thinking about other work by some of the authors here, specifically Háden's 2024 paper in Cognition suggesting that newborn EEG responses to beats do NOT reflect statistical learning. How do these two pieces of work complement each other (or not)?

- Last paragraph of discussion: Social interactions do not begin at 6 months

- The links to language presented in the end of the discussion are unclear. I'd argue that speech processing usually relies on temporal cues more than pitch cues unless we are thinking about prosody. But for understanding word meaning and basic communication, why would pith be prioritized?

Reviewer #3 (Anne Kösem): This study utilizes EEG recordings in newborns to examine whether infants can track pitch and rhythmic expectations in musical excerpts. The authors found that encoding of probabilistic rhythmic expectations occurred only in response to real music, as opposed to shuffled music. However, no evidence was observed regarding the tracking of melodic information.

I would like to express my appreciation for the clarity and quality of the manuscript. The study presents a highly relevant topic and is executed with great competence. Furthermore, the methodologies employed are robust and well-founded.

My main inquiry pertains to the interpretation of the results related to rhythmic expectations. I am curious if these findings genuinely support the existence of predictive timing mechanisms as the authors suggest. Alternatively, these results might indicate non-predictive mechanisms, potentially arising from the superposition of evoked responses that arrive at different timings. The authors report a significant effect of inter-onset interval on the mTRF, which, in my view, could signify the impact of the evoked response to the preceding sound—possibly occurring as recently as 100 ms prior. Moreover, it is conceivable that the inter-onset interval of the subsequent tone may also affect the mTRF. If, in real music, the inter-onset interval of the next tone is biased in duration under conditions of high surprise versus low surprise, this could further influence the observed results.

---

## [Decision Letter · Decision Letter 2]

15 Dec 2025

Dear Dr Bianco,

Thank you for your patience while we considered your revised manuscript "Human newborns form musical predictions based on rhythmic but not melodic structure" for publication as a Research Article at PLOS Biology. This revised version of your manuscript has been evaluated by the PLOS Biology editors, the Academic Editor and the original reviewers.

Based on the reviews, we are likely to accept this manuscript for publication, provided you satisfactorily address the remaining points raised by the reviewers Please also make sure to address the following data and other policy-related requests:

* Please add the links to the funding agencies in the Financial Disclosure statement in the manuscript details.

* Please include the approval/license number of the ethical approval for the experiments.

* DATA POLICY:

Regardless of the method selected, please ensure that you provide the individual numerical values that underlie the summary data displayed in the following figure panels as they are essential for readers to assess your analysis and to reproduce it: 2BC, S3A, S4 and S5.

* CODE POLICY

Per journal policy, if you have generated any custom code during the course of this investigation, please make it available without restrictions. Please ensure that the code is sufficiently well documented and reusable, and that your Data Statement in the Editorial Manager submission system accurately describes where your code can be found. [IF APPLICABLE: As the code that you have generated to XXX is important to support the conclusions of your manuscript, its deposition is required for acceptance.]

We expect to receive your revised manuscript within two weeks.

*Published Peer Review History*

*Press*

Sincerely,

Christian

Christian Schnell, PhD

Senior Editor

cschnell@plos.org

PLOS Biology

Reviewer remarks:

Reviewer #1 (Benjamin Morillon): We congratulate the authors on this high-quality work.

Reviewer #2: I thank the authors for their careful and clear revisions and responses to reviewer comments. I feel the manuscripts clarity and impact is improved and I have no further comments.

Reviewer #3 (Anne Kösem): The authors have satisfactorily addressed my concern. I suggest including the response to the reviewers that addresses how larger IOIs are associated with relatively more surprising notes (in music) in the main text (for example, in the results section page 6). I believe this is an important point that is only currently indicated in the legend of Figure S3. I am referring to this particular response:

« A linear mixed model predicting the preceding IOI, with factors condition (real /shuffled) and surprise level (high / low), yielded no main effect of condition (χ2(1) = 1.12, p = 0.29), butsignificant main effect of surprise level (χ2(1) = 16.89, p < .001), and an interaction of condition andsurprise level (χ2(1) = 18.12, p < .001), suggesting that this bias is stronger in real than in shuffled music (Figure S3A, left panel). However, we do not believe that this constitutes an issue for our interpretation of the results because our TRF analyses included the preceding IOI as a regressor, allowing us to separate its unique contribution from that of Musical Surprise. Indeed, the effects attributed to the high level rhythmic tracking (St and Et model) cannot be explainable by low-level timing alone, as the St and Et regressors accounts for additional EEG variance beyond that explained by the preceding IOI »

---

## [Editor Report · Decision Letter 3]

6 Jan 2026

Dear Roberta,

Happy New Year!

Thank you for the submission of your revised Research Article "Human newborns form musical predictions based on rhythmic but not melodic structure" for publication in PLOS Biology and apologies for the delay in getting back to you over the holiday period.

On behalf of my colleagues and the Academic Editor, Mathew Diamond, I am pleased to say that we can in principle accept your manuscript for publication, provided you address any remaining formatting and reporting issues. These will be detailed in an email you should receive within 2-3 business days from our colleagues in the journal operations team; no action is required from you until then. Please note that we will not be able to formally accept your manuscript and schedule it for publication until you have completed any requested changes.

PRESS

Sincerely,

Christian

Christian Schnell, PhD

Senior Editor

PLOS Biology

cschnell@plos.org